# Factors Associated with Traumatic Diaphragmatic Rupture among Patients with Chest or Abdominal Injury: A Nationwide Study from Japan

**DOI:** 10.3390/jcm11154462

**Published:** 2022-07-30

**Authors:** Yusuke Katayama, Kenta Tanaka, Kenichiro Ishida, Tomoya Hirose, Jotaro Tachino, Shunichiro Nakao, Yutaka Umemura, Kosuke Kiyohara, Masahiro Ojima, Takeyuki Kiguchi, Tetsuhisa Kitamura, Jun Oda

**Affiliations:** 1Department of Traumatology and Acute Critical Medicine, Osaka University Graduate School of Medicine, Suita 565-0871, Japan; htomoya1979@hp-emerg.med.osaka-u.ac.jp (T.H.); jotarotachino@gmail.com (J.T.); shunichironakao@gmail.com (S.N.); odajun@gmail.com (J.O.); 2Division of Environmental Medicine and Population Sciences, Department of Social and Environmental Medicine, Osaka University Graduate School of Medicine, Suita 565-0871, Japan; tanaken.0414@gmail.com (K.T.); lucky_unatan@yahoo.co.jp (T.K.); 3Department of Acute Medicine and Critical Care Center, Osaka National Hospital, National Hospital Organization, Osaka 540-0006, Japan; kenichiro1224@gmail.com (K.I.); ojimarionet@yahoo.co.jp (M.O.); 4Department of Emergency and Critical Care, Osaka General Medical Center, Osaka 558-8558, Japan; plum0022@yahoo.co.jp (Y.U.); take_yuki888@yahoo.co.jp (T.K.); 5Department of Food Science, Faculty of Home Economics, Otsuma Women’s University, Tokyo 102-8357, Japan; kiyohara@otsuma.ac.jp

**Keywords:** traumatic diaphragmatic rupture, blunt trauma, abdominal injury, chest injury, epidemiology

## Abstract

Background: Blunt traumatic diaphragmatic rupture (TDR) is a rare condition that is seen in patients with blunt thoracoabdominal trauma. However, factors that are associated with blunt TDR have not been fully revealed. The purpose of this study was to evaluate the factors that are associated with blunt TDR in trauma patients with a chest or abdominal injury using nationwide trauma registry data in Japan. Method: This study was a retrospective observational study with a 15-year study period from 2004 to 2018. We included trauma patients with a chest or abdominal Abbreviated Injury Score of two or more. We evaluated the relationship between confounding factors such as mechanism of injury and blunt TDR with multivariable logistic regression analysis. Results: This study included 65,110 patients, of whom 496 patients (0.8%) suffered blunt TDR. Factors that were associated with blunt TDR were disturbance of consciousness (adjusted OR [AOR]: 1.639, 95% CI: 1.326–2.026), FAST positive (AOR: 2.120, 95% CI: 1.751–2.567), front seat passenger (AOR: 1.748, 95% CI: 1.129–2.706), and compression injury by heavy object (AOR: 1.677, 95% CI: 1.017–2.765). Conclusion: This study revealed several factors that are associated with blunt TDR. The results of this study may be useful for clinicians when estimating blunt TDR.

## 1. Introduction

Blunt traumatic diaphragmatic rupture (TDR) is a rare condition that is seen in patients with blunt thoracoabdominal trauma. In fact, it is estimated to occur in 0.8–5% of patients due to traffic accidents [1,2,3,4]. A retrospective study of 833,309 patients by the American Association for the Surgery of Trauma in 2012 reported an incidence of TDR of 0.46%, with 67% caused by penetrating injury and 33% by blunt trauma [5]. The most common cause of blunt TDR is car-to-car traffic accidents, followed by bicycle-car traffic accidents. The diagnosis of blunt TDR is difficult as the physical findings are mainly respiratory symptoms, but they are not specific; the traditional diagnostic method of chest x-ray is not sufficient to identify it, and computed tomography scanning may misdiagnose it [5,6].

The pathophysiology of penetrating and blunt TDR is complex. Although the diaphragm is generally injured by perforation with a bullet or blade in penetrating TDR, in blunt TDR, the diaphragm is injured by shearing of the stretched diaphragm, detachment of the muscle attachments, and increased abdominal pressure that exceeds the burst pressure of the diaphragm due to the large energy that is applied to the thorax and abdomen. Therefore, although TDR can be predicted from the direction of the bullet or blade in penetrating TDR, it is difficult to predict the complications of TDR in patients with blunt injury. In addition, the recognition of TDR is delayed in these patients because the high energy can cause other life-threatening trauma such as aortic injury, hepatic injury, and pelvic fracture, and treatment of these life-threatening injury takes precedence. Further, the prevalence of non-operative management of blunt trauma such as that by interventional radiology (IVR) has reduced the opportunity to identify blunt TDR that would have been detected during laparotomy in the past. As a result, the late detection of blunt TDR leading to gastrointestinal herniation is associated with high mortality and morbidity. If the factors that are associated with blunt TDR can be identified, clinicians may be able to recognize blunt TDR earlier, thus leading to improvement in the prognosis of blunt TDR.

The Japanese Trauma Data Bank (JTDB) is a nationwide trauma registry in Japan that is operated by the Japanese Association for the Surgery of Trauma. Data registration began in 2003, and by 2018, approximately 350,000 trauma patients had been registered in this trauma registry [7]. The purpose of this study was to evaluate factors that are associated with blunt TDR in trauma patients with a chest or abdominal injury by analyzing data from this trauma registry.

## 2. Materials and Methods

### 2.1. Study Design and Settings

This study was a retrospective observational study using the JTDB dataset. The study period was the 15 years from 2004 to 2018. We included trauma patients with a chest or an abdominal Abbreviated Injury Score (AIS) of two or more, based on previous studies [8]. We also excluded patients with penetrating injury, patients in whom focused assessment with sonography for trauma (FAST) was not performed, and patients with missing data such as sex and age. This study was approved by the ethics committee of Osaka University Graduate School of Medicine (approval no. 16260). Informed consent from individual patients was waived because the data in the JTDB dataset are anonymized. This manuscript was written in accordance with the STROBE statement [9].

### 2.2. Japanese Trauma Data Bank

We previously described the JTDB in detail [10]. The JTDB was established in 2003 by the Japanese Association for the Surgery of Trauma (Trauma Surgery Committee) and the Japanese Association for Acute Medicine (Committee for Clinical Care Evaluation) [7] and is similar to trauma registries in Europe, Oceania, and the United States. By 2021, the JTDB dataset has been registered by 292 major emergency medical institutions around Japan [11]. These emergency medical institutions have abilities that are equal to those of Level 1 trauma centers in the United States. The data were collected via the Internet from participating institutions. In most cases, the physicians and technicians who attended an AIS coding course registered the data [12].

The JTDB captures data from trauma patients that include age, sex, type of mechanism of injury, AIS code, Injury Severity Score (ISS), vital signs on hospital arrival, data and the time course from hospital arrival to discharge, medical treatments such as neurosurgical treatment and IVR, and complications in accordance with regular forms with coding items [12]. The ISS was calculated from the top three scores of the AIS for nine sites that were classified by AIS codes.

### 2.3. Main Outcome

The main outcome was the occurrence of blunt TDR. We defined blunt TDR according to the AIS codes 440699.2, 440602.2, 440604.3, and 440606.4, which were extracted from the diagnosis codes that were recorded in the JTDB.

### 2.4. Statistical Analysis

We used multivariable logistic regression analysis to evaluate factors that are associated with blunt TDR in patients with chest or abdominal injury and calculated the adjusted odds ratio (AOR) and 95% confidence interval (CI). Multivariable logistic regression analysis was performed using the forced entry method. The potential covariates were age groups (0–14 years, 15–29 years, 30–44 years, 45–59 years, 60–74 years, and 75 years or older), sex, mechanism of injury, disturbance of consciousness, shock, FAST positive, chest AIS score, and abdominal AIS score. The mechanism of injury was categorized as car driver, front seat passenger, back seat passenger, motorcycle rider, pillion passenger, cyclist, pedestrian, injury from height, fall injury from stairs, falling down injury, compression injury by press machine, compression injury by heavy object, injury by falling or flying object, and others. We defined disturbance of consciousness in patients as a Glasgow Coma Scale (GCS) score of less than 8 points at hospital arrival and shock in patients as a shock index greater than one. These covariates for multivariable logistic regression analysis were physiologically essential or factors that previous studies had shown to be associated with blunt TDR [5,8,13]. Statistical significance was defined as *p* < 0.05. We used SPSS statistics ver. 27.0 J (IBM Corp., Armonk, NY, USA) for statistical analysis.

## 3. Results

Figure 1 shows the patient flow in this study. From 2004 to 2018, 353,276 patients were registered in the JTDB, of whom 96,996 patients were those with chest or abdominal injury with a chest or abdominal AIS of two or more. Of these patients, 65,110 were included in this study after the exclusion of patients with penetrating injury (*n* = 2008), patients in whom FAST was not performed (*n* = 10,093), and patients with missing data (GCS: *n* = 3112, shock index: *n* = 12,088, age: *n* = 61, sex: *n* = 18). In total, 496 patients (0.8%) suffered blunt TDR.

Table 1 shows the patient characteristics in this study. The median patient age was 63 years (interquartile range [IQR], 38–76 years), and the most common age group was “60–74 years” (15,922 patients, 24.5%), followed by “45–59 years” (12,672 patients, 19.5%). Among the patients, 46,586 (71.5%) were male, and 9708 patients (14.9%) had disturbance of consciousness. The median ISS was 21 (IQR, 14–29), and the in-hospital mortality rate was 10.5% (6823/62,859 patients). Motorcycle rider was the most common mechanism of injury (11,235 patients; 17.3%), followed by car driver (10,913; 16.8%) and injury from height (10,024; 15.4%).

Table 2 shows the results of multivariable logistic regression analysis. Factors that were associated with blunt TDR were disturbance of consciousness (AOR: 1.362, 95% CI: 1.099–1.686), FAST positivity (AOR: 1.233, 95% CI: 0.994–1.529), front seat passenger (AOR: 1.728, 95% CI: 1.107–2.698), and compression injury by heavy object (AOR: 1.626, 95% CI: 0.978–2.702). In contrast, “0–14 years” (AOR: 0.307, 95% CI: 0.132–0.715), “15–29 years” (AOR: 0.668, 95% CI: 0.476–0.936), shock (AOR: 0.630, 95% CI: 0.512–0.776), injury from height (AOR: 0.590, 95% CI: 0.413–0.842), and fall injury from stairs, etc. (AOR: 0.401, 95% CI: 0.228–0.705) showed an inverse relationship with the occurrence of blunt TDR.

## 4. Discussion

By using the dataset from the JTDB, a national trauma patient registry in Japan, we revealed that disturbance of consciousness, FAST positive, front seat passenger, and compression injury by heavy object were the factors that were associated with blunt TDR. In contrast, younger patients, shock, injury from heights, and fall injury from stairs, etc. were inversely associated with blunt TDR. The factors that were associated with blunt TDR in patients with thoracoabdominal injury that were identified in this study may help clinicians infer blunt TDR, a rare type of trauma, and may contribute to improving outcomes of these patients.

First, front seat passenger was associated with blunt TDR in this study. In a study of patients in a car crash, the incidence of blunt TDR in each seat position was 2.6% for the driver, 1.3% for the front seat passenger, and 1.2% for the back seat passenger [8]. The values in the present study were 0.9%, 1.3%, and 1.2%, respectively, and were similar except for the driver’s seat value. Although it is not clear why the front seat passenger was more highly associated with blunt TDR, previous studies have reported that side forces are more likely to cause blunt TDR than frontal forces [8]. Another factor is that the driver is aware of the impending frontal impact, but the front seat passenger and back seat passenger may not always be aware of the impact, which may cause blunt TDR due to the relationship between breathing and the timing of the impact.

Second, compression injury by heavy object was also associated with blunt TDR. Blunt TDR is conventionally associated with shearing of the stretched diaphragm, detachment of the muscle attachments, and increased abdominal pressure that exceeds the pressure at which the diaphragm bursts, and the results in this study may reflect this pathophysiology. Similarly, it is suggested that compression injury by press machine may also be associated with blunt TDR, but this study did not find a statistically significant difference because it is a rare mechanism of injury. However, this study found that injury from heights and fall injuries from stairs, etc. were inversely associated with blunt TDR. Although the direction of injury in a fall from a height is unknown, it is likely to be from the head or lower extremities in most cases. As a result, the diaphragm may not be injured in a fall from a height because the pressure is not applied to the abdomen at the moment of impact with the ground. Also, fall injury from stairs, etc. would not likely be associated with blunt TDR because no significant pressure would be applied to the abdomen in the first place.

Third, FAST positive was associated with blunt TDR in the present study. The study by Reiff et al. revealed that blunt TDR was associated with pulmonary contusion, rib fractures, thoracic aortic injury, spleen injury, liver injury, and pelvic fracture [8]. In addition, using data from the American College of Surgeons National Trauma Data Bank, Fair et al. revealed higher rates of pulmonary injury, pneumothorax, and spleen injury in blunt TDR compared to penetrating TDR [5]. The result that FAST positivity was associated with blunt TDR may reflect the presence of hemorrhage due to other organ injuries in the chest and abdomen. In this study, we included patients with blunt trauma of the chest and abdomen and consequently also included those with head trauma. This may be one reason why the disturbance of consciousness was associated with blunt TDR. However, shock was not associated with blunt TDR in the present study. Although a clear mechanism for this result is unknown, in several areas of Japan, emergency physicians rush to the scene by ambulance or helicopter to stabilize patients by administering fluids and intubation before hospital arrival [14]. The results indicate that FAST positivity suggesting massive hemorrhage might be associated with blunt TDR, whereas shock did not appear to be. However, this relationship has not been fully revealed and will require further study.

Younger patients were not associated with blunt TDR. Although the reason for this result is unknown, the muscles of the trunk are less developed in younger people than in adults, and the abdominal wall easily extends in response to the sudden increase in abdominal pressure with injury, thereby distributing the pressure applied to the abdomen. As a result, the incidence of blunt TDR may have been low in the young patients because not enough pressure is applied to cause rupture of the diaphragm. However, the detailed mechanism is unknown and will need to be verified with basic research.

There are several limitations in this study. First, it is unclear whether the diagnosis of blunt TDR was based on findings from imaging or those that were revealed during surgical treatment. Second, factors such as seat belt use, air bag activation, and crash speed were not included in this trauma registry and, therefore, could not be evaluated. Third, this is a registry-based study, and the possibility exists that cases in which no laparotomy or thoracotomy was performed may have been overlooked in patients who were actually complicated by blunt TDR. Finally, as this was a retrospective observational study, there may be unknown confounding factors.

## 5. Conclusions

This study revealed that disturbance of consciousness, FAST positivity, front seat passenger, and compression injury by heavy object were associated with blunt TDR. This result may be useful for clinicians when estimating the presence of blunt TDR.

## Figures and Tables

**Figure 1 jcm-11-04462-f001:**
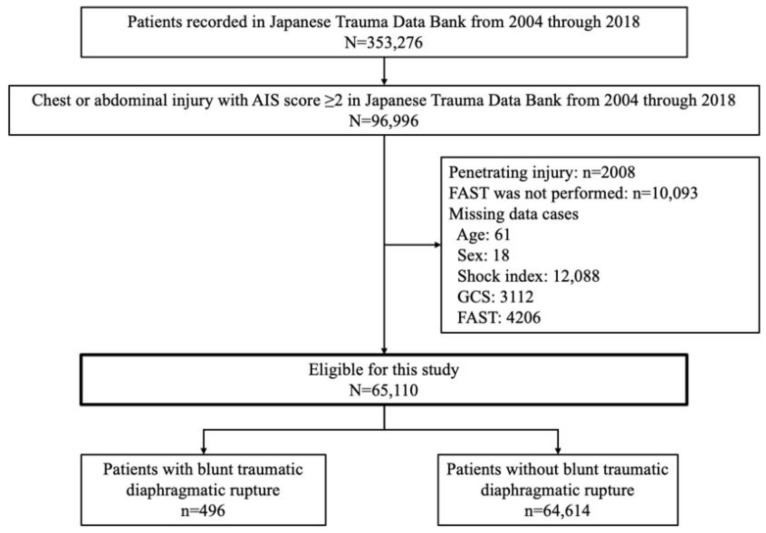
Patient flow in this study. FAST: focused assessment with sonography for trauma; GCS: Glasgow Coma Scale.

**Table 1 jcm-11-04462-t001:** Demographic and clinical characteristics of the study patients with chest or abdominal injury.

Characteristic	Total		Blunt TDR (+)		Blunt TDR (−)	
	(*n* = 65,110)		(*n* = 496)		(*n* = 64,614)	
Age, years, median (IQR)	54	(33–70)	58	(40–71)	54	(32–70)
Age group, years, *n* (%)						
0–14	2704	(4.2)	6	(1.2)	2698	(4.2)
15–29	11,708	(18.0)	12	(12.3)	11,647	(18.0)
30–44	10,686	(16.4)	83	(16.7)	10,603	(16.4)
45–59	12,672	(19.5)	110	(22.2)	12,562	(19.4)
60–74	15,922	(24.5)	145	(29.2)	15,777	(24.4)
≥75	11,418	(17.5)	91	(18.3)	11,327	(17.5)
Male, *n* (%)	46,586	(71.5)	323	(65.1)	46,263	(71.6)
Disturbance of consciousness (GCS < 8), *n* (%)	9708	(14.9)	151	(30.4)	9557	(14.8)
Shock, *n* (%)	53,173	(81.7)	279	(56.3)	52,894	(81.9)
FAST positive, *n* (%)	12,213	(18.8)	193	(38.9)	12,020	(18.6)
Mechanism of injury, *n* (%)						
Traffic accident						
Car driver	10,913	(16.8)	94	(19.0)	10,819	(16.8)
Front seat passenger	2304	(3.5)	31	(6.3)	2304	(3.5)
Back seat passenger	1367	(2.1)	16	(3.2)	1351	(2.1)
Motorcycle rider	11,235	(17.3)	79	(15.9)	11,156	(17.3)
Pillion passenger	383	(0.6)	0	(0)	383	(0.6)
Cyclist	5131	(7.9)	31	(6.3)	5100	(7.9)
Pedestrian	7224	(11.1)	94	(19.0)	7130	(11.1)
Injury from height	10,024	(15.4)	54	(10.9)	9970	(15.4)
Fall injury from stairs, etc.	6968	(10.7)	15	(3.0)	6953	(10.7)
Falling down injury	2544	(3.9)	12	(2.4)	2532	(3.9)
Compression injury by press machine	75	(0.1)	1	(0.2)	74	(0.1)
Compression injury by heavy object	1332	(2.0)	20	(4.0)	1312	(2.0)
Injury by falling or flying object	510	(0.8)	4	(0.8)	506	(0.8)
Others	5100	(7.8)	45	(9.1)	5055	(7.8)
Chest AIS, *n* (%)						
1	851	(1.3)	0	(0)	851	(1.3)
2	6149	(9.4)	12	(2.4)	6137	(9.5)
3	25,625	(39.4)	85	(17.1)	25,540	(39.5)
4	20,102	(30.9)	337	(67.9)	20	(30.6)
5	3636	(5.6)	59	(11.9)	3577	(5.5)
6	149	(0.2)	3	(0.6)	146	(0.2)
None/unknown	8598	(13.2)	0	(0)	8598	(13.3)
Abdominal AIS, *n* (%)						
1	1076	(1.7)	14	(2.8)	1062	(1.6)
2	9280	(14.3)	108	(21.8)	9172	(14.2)
3	8299	(12.7)	92	(18.5)	8207	(12.7)
4	3493	(5.4)	71	(14.3)	3422	(5.3)
5	741	(1.1)	30	(6.0)	711	(1.1)
6	21	(0.0)	1	(0.2)	20	(0.0)
None/unknown	42,200	(64.8)	180	(36.3)	42,020	(65.0)
ISS, median (IQR)	21	(14–29)	32	(24–41)	21	(14–29)
In-hospital mortality, *n* (%)	6823	(10.9)	122	(26.2)	6701	(10.7)

AIS: Abbreviated Injury Scale; DR: diaphragmatic rupture; FAST: focused assessment with sonography for trauma; GCS: Glasgow Coma Scale; IQR: interquartile range; ISS: Injury Severity Score; TDR: traumatic diaphragmatic rupture.

**Table 2 jcm-11-04462-t002:** Factors that were associated with traumatic diaphragmatic rupture.

	Proportion % (*n*/*N*)		Adjusted OR (95% CI)		*p* Value
Age group, years					
0–14	0.2	(6/2704)	0.307	(0.132–0.715)	0.006
15–29	0.5	(61/11,708)	0.668	(0.476–0.936)	0.019
30–44	0.8	(83/10,686)	Reference		
45–59	0.9	(110/12,672)	1.212	(0.905–1.623)	0.197
60–74	0.9	(145/15,922)	1.258	(0.950–1.666)	0.110
≥75	0.8	(91/11,418)	0.989	(0.720–1.358)	0.944
Sex					
Male	0.7	(323/46,586)	Reference		
Female	0.9	(173/18,524)	1.177	(0.961–1.442)	0.115
Disturbance of consciousness					
GCS < 8	1.6	(151/9708)	1.362	(1.099–1.686)	0.005
GCS ≥ 8	0.6	(345/55,402)	Reference		
Shock					
Shock (+)	0.5	(279/53,173)	0.630	(0.512–0.776)	<0.001
Shock (−)	1.8	(217/11,937)	Reference		
FAST					
Positive	1.6	(193/12,213)	1.233	(0.994–1.529)	0.057
Negative	0.6	(303/52,897)	Reference		
Chest AIS (every one score up)			2.079	(1.896–2.278)	<0.001
Abdominal AIS (every one score up)			1.520	(1.431–1.615)	<0.001
Mechanism of injury					
Traffic accident					
Car driver	0.9	(94/10,913)	1.184	(0.869–1.615)	0.285
Front seat passenger	1.3	(31/2304)	1.728	(1.107–2.698)	0.016
Back seat passenger	1.2	(16/1367)	1.476	(0.840–2.593)	0.176
Motorcycle rider	0.7	(79/11,235)	Reference		
Pillion passenger	0.0	(0/383)	-	-	-
Cyclist	0.6	(31/5131)	1.355	(0.974–1.885)	0.071
Pedestrian	1.3	(94/7224)	0.955	(0.620–1.470)	0.835
Injury from height	0.5	(54/10,024)	0.590	(0.413–0.842)	0.004
Fall injury from stairs, etc.	0.2	(15/6968)	0.401	(0.228–0.705)	0.002
Falling down injury	0.5	(12/2544)	1.055	(0.563–1.975)	0.868
Compression injury by press machine	1.3	(1/75)	2.084	(0.282–15.419)	0.472
Compression injury by heavy object	1.5	(20/1332)	1.626	(0.978–2.702)	0.061
Injury by falling or flying object	0.8	(4/510)	0.935	(0.336–2.606)	0.898
Others	0.9	(45/5100)	1.193	(0.819–1.737)	0.358

AIS: Abbreviated Injury Scale; CI: confidence interval; FAST: focused assessment with sonography for trauma; GCS: Glasgow Coma Scale; OR: odds ratio.

## Data Availability

The data that support the findings of this study are available from the JTDB, but the availability of these data is restricted.

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
