# Peer review of "Factors Associated with Traumatic Diaphragmatic Rupture among Patients with Chest or Abdominal Injury: A Nationwide Study from Japan"

_jcm, 2022, doi:10.3390/jcm11154462_

Round 1
Reviewer 1 Report
Thanks for the opportunity to review this retrospective work on traumatic diaphragmatic hernia from a national data base.
I ha d a few queries
1.Does the data base have information on imaging studies performed? If it does why that has not been studied.
2. Why a GCS< 8 as criteria for disturbed consciousness. GCS< 8 the patient is usually intubated and difficult to assess. Is that the reason?
3. I would suggest studying all patents regardless of FAST status as FAST is for identifying free fluid and has nor role in diagnosing diaphragmatic injuries.
4. What are the seat belt regulations in Japan? Does this have any implication for the injury pattern?
Author Response
Thanks for the opportunity to review this retrospective work on traumatic diaphragmatic hernia from a national data base.
I had a few queries
Thank you for your throughout review and valuable comments. Our responses to your queries as follows:
1.Does the data base have information on imaging studies performed? If it does why that has not been studied.
Thank you for pointing this out. Data on imaging studies, and of the 65,110 patients in this study, 58,187 patients (89.4%) had chest CT scanning and 56,499 patients (86.8%) had abdominal CT scanning. On the other hand, 8,590 patients in which emergency open thoracotomy or laparotomy was performed had CT scanning in all cases. Therefore, it is unclear whether the diagnosis of blunt TDR was based on imaging studies or intraoperative findings, so data about imaging studies was excluded from the analysis. This point was described in Limitation section as follows.
“First, it is unclear whether the diagnosis of blunt TDR was based on findings from imaging or those revealed during surgical treatment.”
- Why a GCS< 8 as criteria for disturbed consciousness. GCS< 8 the patient is usually intubated and difficult to assess. Is that the reason?
As you pointed out, it is difficult to assess the level of consciousness when the patient has a GCS of less than 8, because the patient is usually intubated and anesthetics are used to sedate him or her in some cases. Therefore, a GCS of less than 8 points was used as the definition of disturbance of consciousness in this study.
- I would suggest studying all patents regardless of FAST status as FAST is for identifying free fluid and has nor role in diagnosing diaphragmatic injuries.
Thank you for pointing out this important point. Previous studies have revealed that high energy applied to the chest or abdomen can cause the diaphragm to rupture. However, it is unclear how much pressure is actually applied to the abdomen. Therefore, in this study, we assumed that high energy applied to the abdomen can cause abdominal organ injury and used FAST positivity that is an indicator of abdominal organ injury as an indicator of high energy applied to the abdomen as a covariate.
- What are the seat belt regulations in Japan? Does this have any implication for the injury pattern?
In Japan, it is mandatory to wear a seatbelt when car driving. However, data on whether or not the driver and passengers actually wore seatbelts did not exist in the JTDB dataset, and thus we were not able to evaluate the relationship between blunt TDR and seatbelt wearing. This point was also described in Limitation section as follows.
“Second, factors such as seat belt use, air bag activation, and crash speed were not included in this trauma registry and therefore could not be evaluated.”
Reviewer 2 Report
It is a well done study and I congratulate the authors for their efforts. I'd only recommend them to check their own Japanese association of trauma as is mentioned twice in the manuscript but in different fashion both times.
Author Response
Reviewer 2
It is a well done study and I congratulate the authors for their efforts. I'd only recommend them to check their own Japanese association of trauma as is mentioned twice in the manuscript but in different fashion both times.
Thank you for your throughout review. And I apologized this mistake and revised these words as “the Japanese Association for the Surgery of Trauma”.